**Data Availability Statement:** The data underlying the results presented in the study are available from the Kantar Group (URL: https://www.kantarmedia.com/us). Researchers who wish to access the televised e-cigarette advertising TRPs

# Exposure to e-cigarette TV advertisements among U.S. youth and adults, 2013–2019

**Zongshuan Duan**[1☯], **Yu Wang**[1☯], **Sherry L. Emery**[2], **Frank J. Chaloupka**[3], **Yoonsang Kim**[2], **Jidong Huang**[1]*

**1** School of Public Health, Georgia State University, Atlanta, Georgia, United States of America, **2** NORC at the University of Chicago, Chicago, Illinois, United States of America, **3** Institute for Health Research and Policy, University of Illinois at Chicago, Chicago, Illinois, United States of America

☯ These authors contributed equally to this work.

* jhuang17@gsu.edu

## Abstract

### Introduction

E-cigarette advertising has been shown to increase e-cigarette awareness and use. Although e-cigarette marketing in the early 2010s has been well-documented, little is known about how it has changed in recent years in response to the regulatory scrutiny from the FDA and the Congress to combat youth vaping epidemic. This study aims to examine the exposure to e-cigarette TV advertising among youth and adults in the U.S. from 2013 to 2019, overall and by media market and brand.

### Methods

Quarterly data on e-cigarette TV advertising exposure, measured by target rating points (TRPs), and expenditures from 2013 to 2019 were compiled from the Stradegy™ of Kantar Media. Trends of quarterly e-cigarette advertising TRPs were reported by age group, market, and brand.

### Results

Over the study period, overall exposure to e-cigarette TV advertising was higher among adults than among youth. E-cigarette advertising TRPs and expenditures were relatively stable, despite intermittent fluctuations, between 2013 Q1 and 2017 Q1 except for a one-time dip in 2015 Q3, followed by a sharp decline in 2017 Q2 and stayed low till the end of 2018. A resurgence of e-cigarette advertising TRPs occurred in 2019 Q1, led by the advertising from JUUL, Blu, and Vuse, which peaked in the third quarter of 2019, with quarterly TRPs reaching 316.8 for youth and 1,701.9 for adults, and quarterly advertising expenditure totaling $31 million.

### Conclusions

Significant variations, both over time and across media markets and brands, were observed for e-cigarette televised advertising between 2013 and 2019. Following a lull in TV advertising in 2017/18, major e-cigarette companies have substantially increased advertising of

data need to pay a fee for accessing those data from Kantar Group through a data use license and a data use agreement with Kantar Group. The authors of this study did not receive any special privileges in accessing the data that other researchers would not have.

**Funding:** This study was supported by the National Institutes of Health (grant number R01CA194681, PI Jidong Huang). The funding agencies played no role in study design; in the collection, analysis and interpretation of data; in the writing of the report; and in the decision to submit the article for publication. The content in this paper is solely the responsibility of the authors and does not necessarily represent the official views of the sponsors.

**Competing interests:** The authors have declared that no competing interests exist.

their products on American television since early 2019, resulting in a surge in exposure to e-cigarette advertising among both youth and adults. Our findings highlighted the importance of continued monitoring of e-cigarette advertising in the U.S.

## Introduction

Awareness and use of e-cigarettes have grown substantially among U.S. youth and young adults since e-cigarettes entered the U.S. market around 2006/7 [1–3]. In 2020, 19.6% of high school students and 4.7% of middle school students reported using e-cigarettes in the past 30 days [4]. Studies indicated that increased use of e-cigarettes was, at least partially, attributed to increased e-cigarette awareness, which was influenced by e-cigarette marketing [5–9]. Unlike the advertising of combustible cigarettes, which had been banned on television and radio since 1971, e-cigarette advertising had been largely unregulated in the U.S. until very recently [1]. In 2016, approximately 80% of middle and high school students reported seeing e-cigarette advertising. The primary resources of exposure included retail stores, the Internet, television, and printed media [10].

The e-cigarette marketing strategies have evolved over time. Previous studies documented that total e-cigarette advertising expenditures on television, radio, print, Internet, and outdoors increased exponentially from 2011 to 2013/14 [11, 12], a period when this industry had consolidated into a few major brands either directly owned by the tobacco industry (e.g., Altria's MarkTen) or acquired through mergers, acquisitions, partnerships, or other agreements with existing e-cigarette companies [1]. The early e-cigarette advertising occurred primarily on televisions and print media, using strategies borrowed from the marketing of conventional cigarette brands [13]. Starting in mid-2015, the emergence of JUUL e-cigarettes, a pod-based product resembling a USB flash drive, started a new transformation in the U.S. e-cigarette market [14]. JUUL's success was at least partially attributable to its multiple successful social media marketing campaigns [15, 16].

JUUL's early advertising was criticized for targeting youth by using content appealing to youth in their social media campaigns [16, 17]. In response to the youth vaping epidemic in the U.S., declared by the Surgeon General in 2018, several federal agencies have taken steps regulating and restricting e-cigarette marketing [18]. For example, beginning in April 2018, the U.S. Food and Drug Administration (FDA) and the Federal Trade Commission (FTC) issued multiple warning letters to e-cigarette manufacturers, distributors, and retailers for promoting e-cigarette products in ways misleading to youth or selling e-cigarette products to youth illegally [19–21]. In September 2018, the FDA expanded its "The Real Cost" anti-tobacco campaign to target youth vaping and encouraged e-cigarette manufacturers to voluntarily take actions to prevent youth access to their products [22]. In November 2018, the FDA Commissioner Gottlieb proposed additional steps to prevent youth access to flavored tobacco products [23]. Pressed by investigations conducted by federal and state agencies, JUUL shut down its accounts on Facebook and Instagram, suspended most of its advertising activities, and withdrew flavored (except for menthol) JUULpods from the U.S. retail market in late 2018 and early 2019 [24, 25].

In part as a result of the FDA's intense regulatory scrutiny, major e-cigarette companies altered their marketing strategy by reducing advertising on social media and increasing spending on TV advertising campaigns [24, 26–28]. In January 2019, JUUL launched its "Make the switch" marketing campaign, claiming to target adult cigarette smokers [28]. In their TV commercials, adult smokers shared various reasons why they switched to JUUL and their feelings

after the switch [28]. Two competing brands, Blu and Vuse, followed the footsteps of JUUL and started advertising their own e-cigarette products on television. In March 2019, Reynolds launched a 30-second TV commercial, named "Innovation", to promote Vuse Alto, which highlighted that it was time for innovation to change smoking and Vuse Alto provided these innovations smokers desired. In August, Vuse Alto was further promoted by featuring the "best price" in its TV commercial and each Vuse Alto device was sold only for 99 cents in participating retail stores [22, 27, 29]. From 2019 Q3, Blu e-cigarettes also launched its "Satisfaction" television ads to promote MyBlu device on TV featuring quality, simplicity, and real nicotine satisfaction [23].

Given the strength of the evidence on the relationship between e-cigarette advertising and use of this product, it is important to examine this strategic shift in e-cigarette marketing in the context of a rapidly changing regulatory environment and a rapidly changing media environment in the U.S. Unfortunately, existing studies of e-cigarette TV advertising have so far focused on the pre-2015 period, no studies have specifically examined televised e-cigarette advertising in the U.S. since 2015, a period when substantial changes occurred in the e-cigarette market, including the emergence of JUUL and other pod-based e-cigarettes. Duke *et al.* (2014) examined e-cigarette TV advertising exposure among youth and young adults in the U.S. from 2011 and 2013 using the Nielsen data [30]. They found that exposure to e-cigarette TV advertisements increased 256% among youth and 321% among young adults from 2011 to 2013 in the U.S. [30]. Tuchman (2019) examined the effect of e-cigarette TV advertising on the demand for combustible cigarettes using the Nielsen data from 2009 to 2015, but she did not report overall and market-level TV advertising exposure in her study [31]. In addition, previous studies either focused solely on e-cigarette TV ratings or on marketing expenditures, our study aims to present the data on both TV ratings and TV advertising expenditures. Furthermore, previous studies documented the aggregated e-cigarette TV advertising in the U.S. and did not examine the differences in e-cigarette advertising by media market/geolocation. This study filled these critical research gaps by systematically examining e-cigarette TV advertising exposure by age group, market, and product brand, from 2013 to 2019, which provided a more detailed and in-depth understanding of the televised e-cigarette advertising in the U.S.

## Materials and methods

### Data

The data on e-cigarette TV advertising and marketing expenditures were compiled from the Stradegy™ of Kantar Media. Kantar Media is a data and consulting company that tracks advertising and marketing expenditures at product level for more than 3 million brands. Kantar Media tracks 20 media channels, including TV (network TV, spot TV, cable TV, etc.), print media, radio, online, mobile, video, etc. [32]. In this study, we analyzed e-cigarette-specific advertising data on all TV channels. E-cigarette TV advertising data were retrieved using a list of e-cigarette keywords, which was compiled from previous studies, and updated with additional keywords of new e-cigarette brands generated through Google searches, and searches on social media platforms such as Twitter, Instagram, YouTube, and Reddit [33, 34]. This list of e-cigarette keywords (see S2 Table), which included both generic terms for e-cigarettes (electronic cigarette, e-cig, ecig) and their components (e.g., cartridge, pod, e-juice), and slang terms (e.g., vape, vapor, vaping), was used to retrieve the e-cigarette TV advertising through Kantar Media's search portal. Importantly, we also manually reviewed all products under Kantar product categories that we deemed likely to include e-cigarette products, such as "smoking materials and accessories" [34]. E-cigarette products identified from Kantar were carefully reviewed, and irrelevant products, such as "Nike Vapor", were excluded from data analysis.

Products under the same brand were grouped together (e.g., Vuse Alto, Vuse Ciro, Vuse Solo, and Vuse Vibe were all categorized under the Vuse brand). Our search of Kantar advertising data using this method yielded 31 unique e-cigarette brands and 259 vape shops. The TV advertising and expenditures data for these 31 brands and 259 vape shops for the period 2013 Q1 to 2019 Q4 were retrieved, compiled, and analyzed.

### Measures

Quarterly target rating points (TRPs) were used to assess potential exposure to e-cigarette TV advertising among youth (aged 12–17) and adults (aged 18 years old). The TRPs were calculated as the multiplication of the percentage of people exposed to an advertisement and the average number of times this advertisement was seen by these viewers [35]. Take an advertisement with TRPs = 25 among youth in 2013 Q1 for an example, this ad was likely to have been viewed for an average of 5 times by 5% of youth, or an average of 1 time by 25% of youth in 2013 Q1. It should be noted that the TRPs is an aggregated measure of TV ratings in a media market, which may not represent the actual advertising exposure at the individual level.

Quarterly (from 2013 Q1 to 2019 Q4) televised advertising expenditure was defined as the estimated dollar amount spent by the e-cigarette companies to purchase advertising space on TV [32]. The quarterly e-cigarette TV advertising expenditure was summarized in the unit of $1,000 U.S. dollars without adjustment for inflation.

### Analysis

Data management and analyses were conducted using Stata 15.0 (College Station, TX: Stata-Corp LLC.). We reported the quarterly TRPs of youth and adults, and the quarterly TV advertising expenditures. In addition, we identified the most advertised e-cigarette brands based on their total quarterly household gross rating points (GRPs, a measure of household TV ratings similar to TRPs without specific targeted population) from 2013 to 2019, and we calculated the market-level TRPs for youth and adults for all 210 media markets defined by Kantar Media. In this study, we reported the trends of quarterly TRPs by age for 10 media markets with the highest overall quarterly TRPs. Furthermore, we reported the trends of quarterly TRPs for seven most advertised e-cigarette brands by age over the study period.

## Results

### E-cigarette TV advertising TRPs and expenditures

Fig 1 displays the trends of quarterly e-cigarette TRPs among youth and adults and TV advertising expenditures (in $1,000) from 2013 Q1 to 2019 Q4. Over the study period, e-cigarette advertising TRPs for youth and adults were strongly correlated with the TV advertising expenditures (Pearson correlation = 0.60 for youth and = 0.94 for adults). The e-cigarette advertising TRPs among adults were consistently higher than those of youth. Despite intermittent fluctuations, e-cigarette advertising TRPs and expenditures were relatively stable between 2013 Q1 and 2017 Q1 except for a one-time dip in 2015 Q3. E-cigarette TV advertising were almost non-existent between 2017 Q2 and 2018 Q4, with only a few ads airing during the four quarters from 2017 Q2 to 2018 Q1. Both the youth and adult TRPs increased substantially in 2019 Q1 (from zero ratings to 100.8 among youth and to 528.1 among adults). E-cigarette marketing expenditure also reached to a new high of $12,120,900 in 2019 Q1. Adult TRPs peaked in 2019 Q3 (1,701.9), accompanied by the highest quarterly advertising expenditure ($31,287,100). In 2019 Q4, the expenditure dropped to $6,531,700, and TRPs also declined in both groups (64.9 for youth and 351.6 for adults). Detailed information was presented in S2 Table.

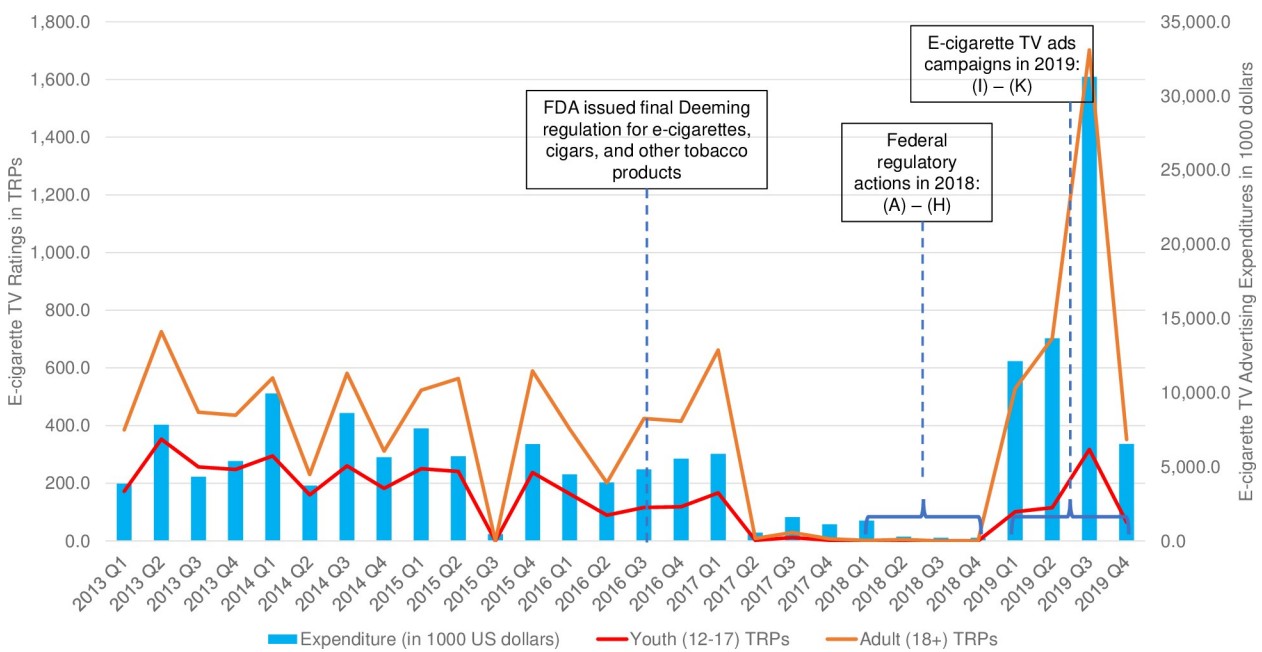

**Fig 1. Quarterly e-cigarette TV advertising expenditures and TV target rating points (TRPs) among U.S. youth and adults, 2013–2019.** (A) February 24, 2018, FDA sent a letter to JUUL expressing concerns about the popularity of JUUL products among youth and requested documents regarding its marketing practices. (B) April 24, 2018, the FDA Commissioner announced new enforcement actions and a Youth Tobacco Prevention Plan to stop youth use of, and access to, JUUL and other e-cigarettes. (C) May 1, 2018, FDA and FTC took actions against companies misleading kids with e-liquids that resembled children's juice boxes, candies and cookies. (D) June 2018, JUUL announced it would "no longer use models on social media platforms," instead focusing on testimonials from adult smokers who switched to JUUL. (E) September 12, 2018, FDA declared youth vaping as "an epidemic proportion" and put makers of the most popular e-cigarette devices on notice that they need to prove they can keep their devices away from minors. (F) September 18, 2018, FDA launched an anti-vaping media campaign. (G) November 15, 2018, the FDA Commissioner proposed a new plan to protect youth by preventing youth access to flavored tobacco products. (H) November 29, 2018, FDA warned companies for selling e-liquids that resemble kid-friendly foods as part of the agency's ongoing Youth Tobacco Prevention Plan. (I) January 2019, JUUL launched its "Make the switch" marketing campaign. (J) March 2019, Reynolds launched a TV commercial, named "Innovation", to promote Vuse Alto e-cigarettes. (K) August 2019, Blu launched its "Satisfaction" television ads to promote MyBlu e-cigarettes.

## E-cigarette advertising TRPs in the top 10 media markets

We identified the top 10, out of 210, media markets with the highest cumulated TRPs from 2013 to 2019. Those top 10 media markets for youth, ranked from high to low, were: Oklahoma City (OK), Tulsa (OK), Pittsburgh (PA), Miami (FL), Buffalo (NY), Tri Cities (WA), Richmond (VA), Toledo (OH), Colorado Springs (CO), and Greenville (SC). The top 10 media markets for adults, ranked from high to low, were: Tulsa (OK), Oklahoma City (OK), Pittsburgh (PA), Colorado Springs (CO), Richmond (VA), Omaha (NE), Lexington (KY), Buffalo (NY), Miami (FL), and Sioux City (IA).

Figs 2 and 3 presented the trends of quarterly e-cigarette advertising TRPs in these 10 media markets, separately by youth and adults. For youth, Richmond was the media market with the highest TRPs (305.5) in 2013 Q1. Miami was the media market with highest TRPs in 2013 Q2 (600.1), Q3 (457.7) and 2014 Q1 (669.4). In 2013 Q4, Tulsa was the most advertised media market with TRPs of 869.5. Tulsa was also the market with the highest TRPs between 2014 Q2-2015 Q1. Oklahoma City had the highest TRPs between 2016 Q1 and 2017 Q4, during which the peak of youth TRPs occurred in 2017 Q1(767.4). Toledo had the highest TRPs in 2018 Q1 (341.6) and Q4 (80.8) and 2019 Q1 (310.0). Youth TRPs in these 10 markets did not differ substantially in the period of 2019 Q2-Q4. For adults, Richmond was the market with the highest TRPs in 2013 Q1(753.6) and Q2 (1,424.8). Miami was the market with the highest

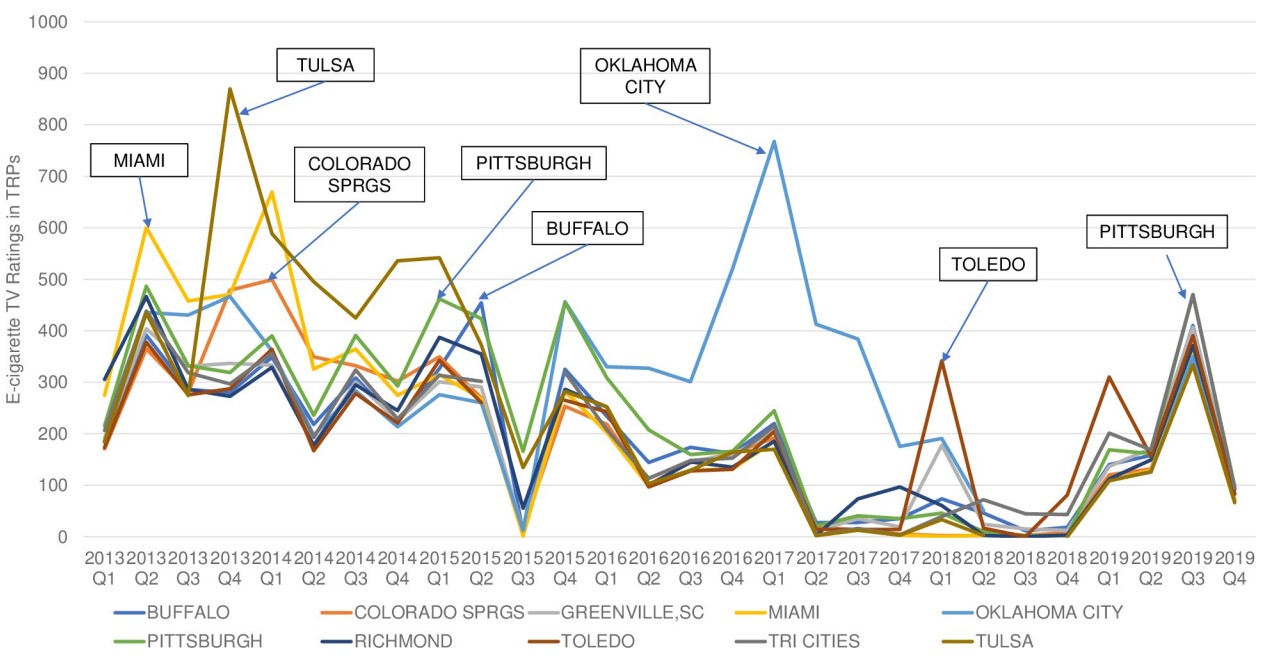

**Fig 2. Youth (age 12–17) quarterly e-cigarette TV target rating points (TRPs) by top 10 Nielsen retail market, 2013–2019.**

TRPs in 2013 Q3 (1,198.2). From 2013 Q4 to 2015 Q4, Tulsa was the market with the highest e-cigarette advertising TRPs, with the peak occurred in 2014 Q4 (2,890.4). Oklahoma City had the highest TRPs between 2016 Q1 to 2018 Q1, during which the highest TRPs occurred in 2017 Q1 (2,033.5). Omaha had the highest quarterly TRPs from 2018 Q2 to 2019 Q3, during which the maximum TRPs were observed in in 2019 Q3 (2,461.3).

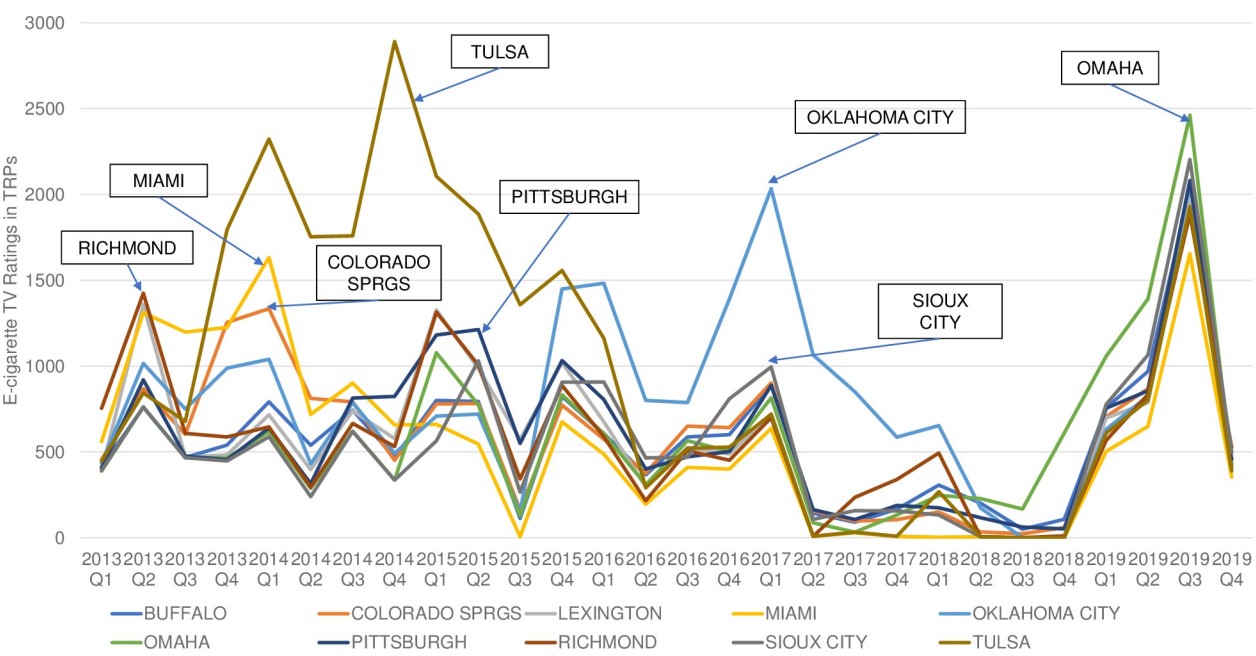

**Fig 3. Adult (age 18 and above) quarterly e-cigarette TV target rating points (TRPs) by top 10 Nielsen retail market, 2013–2019.**

**Table 1. TV ratings overall and by age groups and televised advertising expenditures of the top 20 advertised e-cigarette brands, 2013 to 2019.**

| Products | Youth (aged 12–17) TRPs | Adult (aged 18+) TRPs | Household GRPs | Expenditure (in 1,000 US dollar) |
|---|---|---|---|---|
| **Blu** | 1,862.4 | 3,914.6 | 7,036.7 | 46,373.6 |
| **Vuse** | 1,172.3 | 3,484.0 | 6,185.4 | 43,384.4 |
| **Logic** | 340.1 | 1,335.8 | 2,366.3 | 10,679.0 |
| **JUUL** | 223.2 | 1,264.9 | 2,207.0 | 29,995.5 |
| **Njoy** | 154.6 | 321.3 | 552.7 | 6,308.5 |
| **O2Pur** | 105.1 | 294.0 | 529.1 | 9,951.6 |
| **Fin** | 43.8 | 126.0 | 225.5 | 1,339.5 |
| **21st Century Smoke** | 13.4 | 19.4 | 33.2 | 339.4 |
| **Starfire** | 4.6 | 8.0 | 15.0 | 776.3 |
| **Cue** | 3.0 | 7.1 | 13.0 | 216.2 |
| **Vchic** | 1.8 | 2.6 | 5.1 | 128.8 |
| **Square** | 1.2 | 2.4 | 4.4 | 32.3 |
| **VaporX** | 0.1 | 0.6 | 1.1 | 45.0 |
| **Vero** | 0.0 | 0.1 | 0.3 | 4.7 |
| **EsmokeUSA** | 0.2 | 0.1 | 0.3 | 8.1 |
| **Magic Mist** | 0.0 | 0.0 | 0.0 | 1.7 |
| **Grand Total** | **3,926.6** | **10,783.4** | **19,179.7** | **149,667.7** |

### E-cigarette advertising TRPs for the top seven brands

Based on the household GRPs from 2013 Q1 to 2019 Q4, the top seven most advertised e-cigarette brands were: Blu (cumulative household GPRs = 7,037.6), Vuse (6,185.4), Logic (2,366.3), JUUL (2,270.0), Njoy (552.7), O2Pur (529.1), and Fin (225.5) (Table 1).

Figs 4 and 5 displayed the e-cigarette advertising youth and adult TRPs for the top 7 brands from 2013 Q1 to 2019 Q4. Over the study period, although adult e-cigarette advertising TRPs

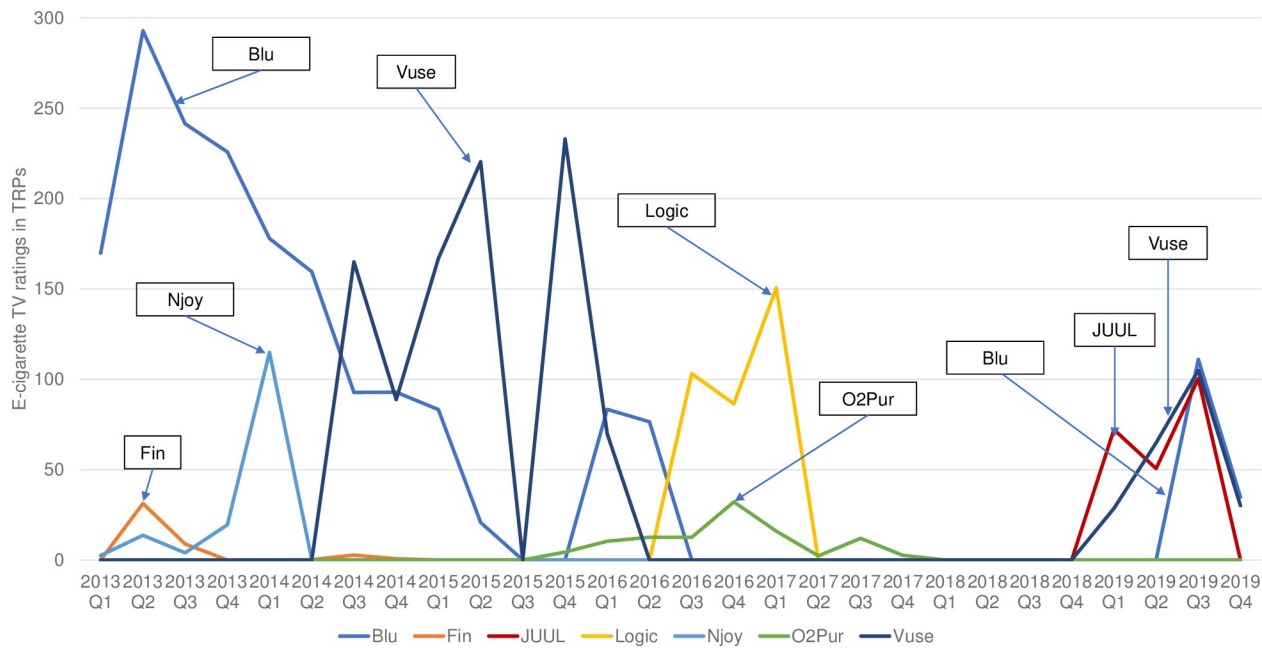

**Fig 4. Youth (age 12–17) quarterly TV target rating points (TRPs) of the top 7 e-cigarette brands, 2013–2019.**

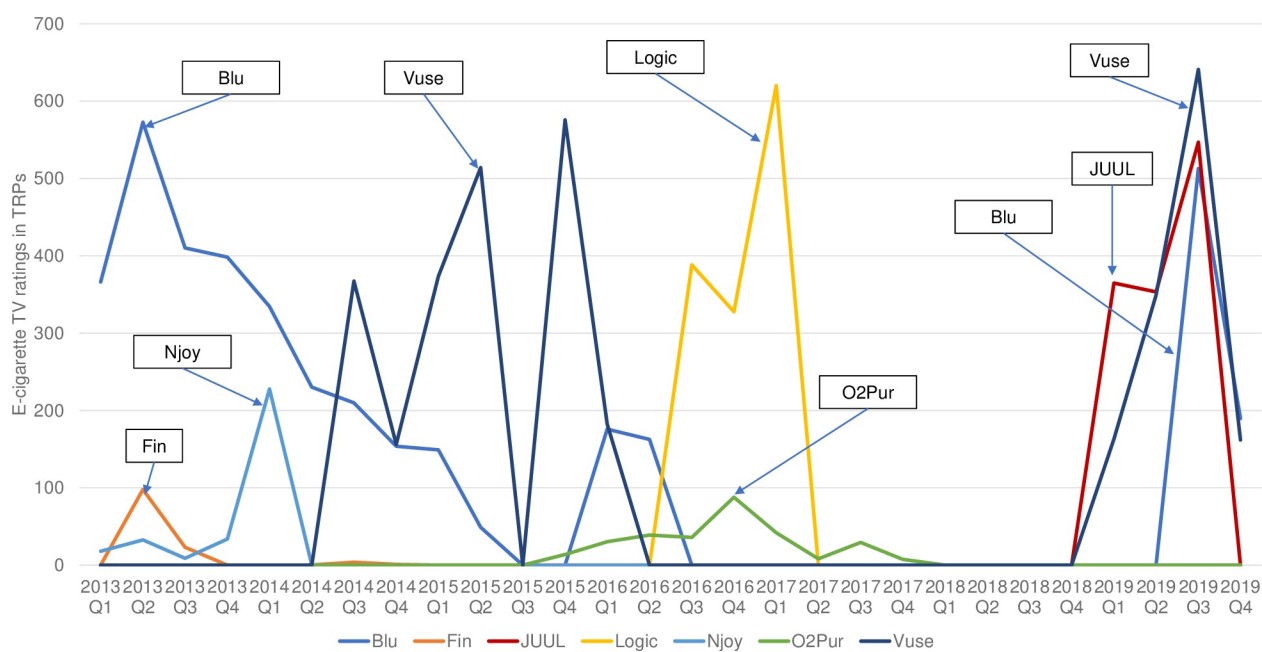

**Fig 5. Adult (aged 18 and above) quarterly TV target rating points (TRPs) of the top 7 e-cigarette brands, 2013–2019.**

were consistently higher than those of youth, the time trends of TRPs for the two groups were similar. In 2013, Blu was the most advertised e-cigarette brand, measured by its quarterly TRPs, which reached the highest levels in 2013 Q2 (292.8 for youth and 572.6 for adults). Vuse overtook Blu in 2014 Q3, becoming the most advertised brand until 2015 Q4. The peak of Vuse TV advertising occurred in 2015 Q4 (233.0 for youth and 575.6 for adults). From 2016 Q3 to 2017 Q1, Logic's ads were the most watched e-cigarette TV ads among U.S. youth and adults, with the highest TRPs occurring in 2017 Q1 (150.5 for youth and 619.9 for adults). Since 2019, the resurgence of e-cigarette ads on the U.S. television was driven entirely by the ads from Blu, JUUL, and Vuse. Both the youth and adult TRPs of these ads peaked in 2019 Q3 (youth: 111.0 for Blu, 100.4 for JUUL, and 104.9 for Vuse; adults: 512.9 for Blu, 546.8 for JUUL, and 640.8 for Vuse).

## Discussion

Our paper provided important insights into the potential e-cigarette TV advertising exposures among American youth and adults since 2013, a period when significant changes, including the emergence of pod-based e-cigarettes such as JUUL, transformed the U.S. e-cigarette industry and marketplace. We found a resurgence in e-cigarette TV advertising since early 2019, after a lull in 2017/2018. Importantly, this resurgence was driven entirely by three major e-cigarette brands, Blu, JUUL, and Vuse. We also observed significant variations of e-cigarette TV advertising exposure across age groups, media markets, and brands.

When e-cigarette products first entered the U.S. market around 2007, they were sold primarily through online retailers with limited marketing [1]. Kornfield and colleagues (2015) did not find marketing expenditure tracked by Kantar Media until 2010 Q1 [34]. The marketing expenditures on e-cigarette TV advertising increased substantially in 2012 when big tobacco companies entered the e-cigarette market [1, 34]. Duke *et al*. (2014) calculated the quarterly TRPs from 2011 to 2013 using the Nielsen data. They found that quarterly youth

TRPs increased from under 100 in the first half of 2012 to 347 in 2013 Q2, consistent with the 353 TRPs reported in our study [30].

Our study extended the previous research by examining the U.S. e-cigarette TV advertising exposure from 2013 through 2019. We found that the e-cigarette advertising TRPs in the U.S. were relatively stable, with intermittent fluctuations, from 2013 Q1 to 2017 Q1 (around 500 TRPs for adults and 200 TRPs for youth), except for a brief dip in 2015 Q3. In addition, our study showed that the average quarterly TV advertising expenditure was approximately $5.0 million from 2013 Q1 to 2017 Q1, consistent with the reports from several previous studies [12, 34]. The substantial marketing spending from 2013 to 2017 likely increased awareness of e-cigarettes and spurred large sales increases [36]. The e-cigarette TV advertising TRPs and expenditures remained low from 2017 Q2 to 2018 Q4, consistent with findings reported by Ali *et al.* (2020). The low level of e-cigarette TV ads during this period likely reflected a shift of e-cigarette companies' marketing strategy from TV, print media and radio to less expensive online/internet advertising [12]. It may also reflect the strategic withdraw of e-cigarette advertising across media channels in response to the increased regulatory scrutiny from federal and state regulators.

After a lull in 2017/18, we found a resurgence in e-cigarette TV advertising in early 2019. Youth TRPs increased from 0.1 in 2018 Q4 to 101 in 2019 Q1, and adult TRPs skyrocketed from 0.3 in 2018 Q4 to 528 in 2019 Q1. E-cigarette advertising TRPs peaked in 2019 Q3 for adults, which reached 1,702. The TV advertising expenditures also reached the peak of more than $30 million in 2019 Q3. These are important findings not reported in previous studies, and represent one of the most important contributions of our study. Notably, the resurgence of e-cigarette TV advertising in 2019 were driven entirely by the ads from three major e-cigarette brands, Blu, JUUL, and Vuse. Previous studies showed the e-cigarette TV advertising expenditure in 2012 was mainly due to the spending of the Blu TV advertising campaigns on the national cable network [11]. Our results also suggested that e-cigarette TV advertising in the U.S. since 2013 was driven by a few major e-cigarette brands, including Vuse's advertising campaign in 2015 and 2016, Logic's advertising in 2017, and the TV advertising campaigns of JUUL, Vuse, and Blu in 2019.

The resurgence in e-cigarette TV advertising in 2019 likely reflected the industry's response to the intense scrutiny from the investigations conducted by several federal agencies and states' Attorney Generals' offices on e-cigarette companies' aggressive marketing on social media. According to a recent Nielsen report, as of 2019 Q2, the average amount of time American adults spent on internet and mobile apps (5:09 hours) has exceeded the average amount of time spent on TV (4:04 hours) per day [37]. Consequently, online advertising and advertising via search engines or mobile apps have become one of the most important ways to reach consumers. Absence of federal and state actions, e-cigarette companies were unlikely to significantly reduce their digital/social media marketing, given the current trend of media consumption.

The resurgence of e-cigarette TV advertising in 2019 may also reflect the industry's strategic shift to focus on adult smokers, as all these three major TV advertising campaigns purportedly aiming at helping adult smokers switch to e-cigarettes. There were approximately 34.1 million adult smokers in the U.S. [38], only a small fraction (approximately 10%) of them were current e-cigarette users [39]. As such, adult smokers represent one of the most important potential markets for e-cigarette companies. For American adults, TV advertising remains one of the most important media channels today [37]. Consequently, the resurgence of e-cigarette TV advertising in 2019 likely reflected e-cigarette companies' strategic decision to promote their products among targeted adult smokers.

In addition, we found that e-cigarette TV advertising TRPs for adults were consistently higher than those for youth during our study period, and the correlation between e-cigarette

TV advertising expenditures and TRPs was stronger for adults than that for youth (Pearson's correlation: 0.94 vs. 0.60). These results indicated the potential exposure of e-cigarette TV advertising was higher among adults than among youth during our study period, and there were potential spillover effects of e-cigarette TV advertising that were aimed at adults. For example, even though the three major e-cigarette TV advertising campaigns in 2019 were all purportedly targeted at adults, youth were exposed to these ads as well, which was evident by the significant increase in TRPs among youth in 2019. Given the evidence on the strong link between e-cigarette advertising exposure and e-cigarette use among youth [1, 9, 40], youth exposure of these TV ads in 2019 could translate into an increase in use of these products.

Our study also revealed significant variations of e-cigarette advertising exposure by media market/geolocations, representing another unique contribution of our study. For example, our results showed that the highest level of youth TV advertising TRPs occurred in the Miami media market in early 2013, followed by the Tulsa media market between late 2013 and early 2015, the Oklahoma City media market between 2016 and 2017, the Toledo media market in 2018, and the Pittsburg media market in 2019. This indicated that the likelihood of seeing an e-cigarette TV ad for youth differs by their location of residence, which could translate into difference in e-cigarette use behaviors.

The findings from our study have several important policy implications: first, the resurgence of televised e-cigarette advertising in 2019 suggests that a more comprehensive e-cigarette advertising regulatory framework may be warranted. Restrictions of youth-oriented e-cigarette advertising should be implemented across all media channels. If restrictions were placed only upon certain media channels, companies can continue promoting their products by shifting to unregulated media channels. Second, given the potential spillover effects of adult-oriented e-cigarette TV advertising, it is important to regulate the timing and location of these ads, and limit the media channels that air such ads. Third, anti-smoking and anti-vaping media campaigns, as well as other location-based tobacco control efforts, conducted by federal and state governments need to take into account the substantial variations in exposure to e-cigarette TV ads by media market/location. For example, in states/media markets where the exposure to e-cigarette TV ads was high, more frequent airing of anti-smoking and ani-vaping media campaigns may be needed. Finally, continued monitoring and surveillance of e-cigarette advertising is needed given the rapidly changing media landscape and the rapidly changing regulatory environment in the U.S.

This study is subject to several limitations. First, TRPs, the measure of potential exposure to e-cigarette TV advertisements, provide an estimate of ads exposure at an aggregated level, and may not represent actual individual level exposure. In addition, our study only reported TV advertising exposure and expenditures. E-cigarette advertising exposure through other channels, such as retail stores, print media, social media, sponsored events, and Internet promotions, were not examined in this study. Future studies can build on our current study to examine the e-cigarette advertising on these other channels.

## Conclusions

Despite these limitations, this study provided important insights into the e-cigarette TV advertising exposures among American youth and adults in recent years, revealing a resurgence in e-cigarette TV advertising in 2019, driven entirely by three major e-cigarette brands. Despite FDA's efforts to combat youth vaping started in 2018, e-cigarette television advertising TRPs and expenditures reached historical high levels in 2019, resulting in an increase in exposure to e-cigarette advertising among both American youth and adults. Given the rapid evolvement of the e-cigarette marketplace and the regulatory environment in the U.S., it is important to

continue monitoring e-cigarettes' advertising and promotional strategies through TV and other media channels, and use data to help guide e-cigarette regulations and policies at the national, state, and local levels.

## Supporting information

**S1 Table. List of keywords to retrieve e-cigarette TV advertising data from Kantar Media.** (DOCX)

**S2 Table. Total quarterly e-cigarette TV advertising expenditure and quarterly e-cigarette TV ratings overall and by age groups, 2013 to 2019.** (DOCX)

## Acknowledgments

The authors would like to thank Glen Szczypka and Steven Binns of NORC for their excellent research assistance.

## Author Contributions

**Conceptualization:** Sherry L. Emery, Frank J. Chaloupka, Yoonsang Kim, Jidong Huang.

**Data curation:** Yoonsang Kim.

**Formal analysis:** Zongshuan Duan, Yu Wang.

**Funding acquisition:** Jidong Huang.

**Investigation:** Yoonsang Kim.

**Methodology:** Zongshuan Duan.

**Supervision:** Frank J. Chaloupka, Jidong Huang.

**Writing – original draft:** Zongshuan Duan, Yu Wang.

**Writing – review & editing:** Yu Wang, Sherry L. Emery, Jidong Huang.

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
