## [Decision Letter · Decision Letter 0]

13 Jan 2021

PONE-D-20-32877

Exposure to e-cigarette TV advertisements among U.S. youth and adults, 2013 – 2019

PLOS ONE

Dear Dr. Huang,

Thank you for submitting your manuscript to PLOS ONE. After careful consideration, we feel that it has merit but does not fully meet PLOS ONE’s publication criteria as it currently stands. Therefore, we invite you to submit a revised version of the manuscript that addresses the points raised during the review process.

Please tighten up the background section and provide a stronger and more detailed interpretation of your results' public health and policy implications.

We look forward to receiving your revised manuscript.

Kind regards,

Stanton A. Glantz

Academic Editor

PLOS ONE

Journal Requirements:

2.We note that you have indicated that data from this study are available upon request. PLOS only allows data to be available upon request if there are legal or ethical restrictions on sharing data publicly. For more information on unacceptable data access restrictions, please see http://journals.plos.org/plosone/s/data-availability#loc-unacceptable-data-access-restrictions.

Reviewers' comments:

Reviewer's Responses to Questions

**Comments to the Author**

1. Is the manuscript technically sound, and do the data support the conclusions?

Reviewer #1: Yes

Reviewer #2: Yes

2. Has the statistical analysis been performed appropriately and rigorously? 

Reviewer #1: Yes

Reviewer #2: N/A

3. Have the authors made all data underlying the findings in their manuscript fully available?

Reviewer #1: Yes

Reviewer #2: Yes

4. Is the manuscript presented in an intelligible fashion and written in standard English?

Reviewer #1: Yes

Reviewer #2: Yes

5. Review Comments to the Author

Reviewer #1: While I appreciate that the authors are reviewing a wide range of time covered by their data, the background section is very long and general. The manuscript would be stronger if the background section was more focused specifically on the issues directly addressed by the analysis.

The background also includes points that are not adequately connected to the paper’s central narrative. For example, it is unclear how the point summarized in the sentence on lines 87-89 demonstrates that federal agencies took a step to regulate e-cigarette marketing in an effort specifically “in response” to youth vaping as stated in the preceding sentence. Do the authors intend that the citation listed for this sentence demonstrates this by providing an example where FDA identified increasing youth vaping rates as motivation for the MRTP approval process and regulation of reduced harm claims? In cases such as this, I suggest the authors revise to make cited support for their argument more clear.

The authors assert on lines 100-101 that “TV remains the most commonly viewed media by youth and adults in the U.S.” with a citation from the Nielsen company dated Quarter 1 2012, almost 9 years ago. Given the rapidly changing media landscape, this assertion should have a more recent citation. This also not a trivial point given the trends that the authors review prior in the section about the increases e-cigarette company direct and indirect digital and social medial advertising driving youth e-cigarette rates. I suggest that the actions of e-cigarette company advertising being directed away from digital/social media back to TV should be contextualized with current information about TV viewership in light of media consumption trends and audience characteristics (with any attendant limitations noted).

Currently, the most directly relevant background information is given relatively short shrift in the paragraph on lines 100-113 (which is only 13 of the 71 lines of text in the section). I think space dedicated to this sort of directly relevant background needs to be better balanced with the disproportionately long preceding account of e-cigarette advertising history.

The sentence on lines 63-65 needs a citation.

Online 130 “Products belonged to the same brand were …” should be edited.

Overall, the approach, data, and methods are sound and consistent with prior studies using similar data that contextualize the results reported here.

In the Discussion section, the authors review the changes to e-cigarette company TV advertising in greater detail than in the background section. I think the paper would be stronger if the review and citations related to the various companys’ 2019 adult-focused ad campaigns was located in the background section to properly contextualize the results. For example, the dates of the various campaign launches across the quarters in 2019 correspond with increased TRPs displayed in the results. After informing the reader of these dates at the outset, the authors might consider adding indicator lines on the key quarters of campaign lunch displayed in figures to call readers’ attention to the corresponding results. Some of the less directly relevant literature reviewed in the background can be edited out to focus the section and accommodate the content.

The manuscript would also be stronger if the authors discussed potential implications of the trends they report beyond the recommendation for continued monitoring. Currently, the conclusions section is quite weak and lessens my enthusiasm for the manuscript. It is important to hear what authors think about their results framed as unintended consequences of FDA scrutiny that motivated changes in e-cigarette company advertising strategies and content. What might the authors expect to see (or not see) if there were continued monitoring? What consequences, for better or worse, do prior studies suggest we may expect from the trends reported here? Some of this is implied in the background and discussion, however the manuscript would be strengthened greatly by spending more space engaging directly in interpretation of results in the discussion section.

Reviewer #2: This study uses consumer media data to examine the exposure to e-cigarette TV advertising between 2013-2019, and more specifically how advertising has changes in response to tobacco regulation. The paper is well-written. The paper would benefit from a greater interpretation of how the findings may or may not relate to specific tobacco regulations enacted over the past 5 years. The Introduction would benefit from a more explicit description of how this study is unique from existing research. This study covers a longer period of observation than previous studies (2013-2019), however it seems other studies have evaluated advertising exposure by age group and evaluated sales by brand. Why is it necessary to measure marketing exposure when sales data are already available? Also, it would help to frame the reasons for studying differences in marketing exposure by market in the Introduction and interpret the results relevant to the market in the Discussion. As written, this analysis seems out of place. Alternatively, the authors might consider dropping the examination of TV advertising exposure by market from this paper.

Introduction

Line 45- update the high school and middle school e-cig prevalence data to reflect the 2020 estimates.

Lines 78-80- wording of sentence is awkward, consider revising. Is the sentence missing a word?

Methods

Line 120- The Kantar Media tracks TV ads across 20 types of media -- what is meant by 20 types of media? Is TV a type of media (as opposed to print/radio) or is this referring to something different? Please clarify.

Line 122- Consider adding the list of keywords as a supplementary files

Page 7 lines 129-130- Please provide more detail as to the process of "careful review" of e-cigarette products identified by Kantar.

Results

Consider condensing the description of Figure 1- focusing on the most salient points. The reader can refer to the figure for details.

The Introduction describes e-cigarette tobacco regulations related to marketing. The figures might benefit from an added overlay identifying when the regulations were enacted. This would allow the reader to easily interpret how regulations may or may not have affected marketing. Table 1 and Figure 1 are redundant-consider moving table 1 to supplementary materials.

Lines 203-207- The description of how the top brands were identified might be better placed in the Methods.

Discussion

The first paragraph of the discussion describes how the findings from this analysis are concordant with that previously reported in the literature- consider starting the paragraph with a concise summary of the unique findings from this paper- those perhaps that have not yet been reported in this paper. Also

Lines 231-238- The description of TV ad expenditure from 2010-2013 seems tangential to the Discussion- would focus more on the paper's findings rather than historical

Figure 1 shows TV ratings for youth and adults- Is this the TRP? Would use the term TRP throughout the Figures rather than rating if this is the case.

Figure 2&3are difficult to read due to many overlying lines- would consider dropping this analysis and Figure from the manuscript or changing the Figures presentation.

6. PLOS authors have the option to publish the peer review history of their article (what does this mean?). If published, this will include your full peer review and any attached files.

Reviewer #1: No

Reviewer #2: No

---

## [Author Response · Author response to Decision Letter 0]

3 Mar 2021

We have revised our manuscript according to the comments of editor and reviewers. Please see below for the point-to-point response to reviewers' comments.

Point-by-point responses to reviewers’ comments:

Reviewer #1: 

Comment #1:

While I appreciate that the authors are reviewing a wide range of time covered by their data, the background section is very long and general. The manuscript would be stronger if the background section was more focused specifically on the issues directly addressed by the analysis.

Response to Comment #1: We appreciate this comment. Per this comment, we’ve shortened the background section, which now focused on articulating the rationale and motivation of our study, addressing issues directly relevant to our analysis. Our study was motivated by several critical gaps in the current literature: first, although previous studies have document televised e-cigarette advertising in the U.S. up to 2013, no studies have specifically examined televised e-cigarette advertising in the U.S. since 2014, a period when substantial changes occurred in the e-cigarette market, including the emergence of JUUL and other pod-based e-cigarettes, and more recent surge in e-cigarette TV advertising by three major brands since early 2019. Second, previous studies either focused solely on e-cigarette TV ratings or on marketing expenditures, our study presented the data on both TV ratings and TV advertising expenditures. Third, previous studies documented the aggregated e-cigarette TV advertising for the U.S. and did not examine the differences in e-cigarette advertising by media market/geolocation, our study filled these critical gaps. In addition, our study examined e-cigarette advertising by age group and by brand, providing a more rich and detailed understanding of the televised e-cigarette advertising in the U.S. 

Comment #2:

The background also includes points that are not adequately connected to the paper’s central narrative. For example, it is unclear how the point summarized in the sentence on lines 87-89 demonstrates that federal agencies took a step to regulate e-cigarette marketing in an effort specifically “in response” to youth vaping as stated in the preceding sentence. Do the authors intend that the citation listed for this sentence demonstrates this by providing an example where FDA identified increasing youth vaping rates as motivation for the MRTP approval process and regulation of reduced harm claims? In cases such as this, I suggest the authors revise to make cited support for their argument more clear.

Response to Comment #2: We appreciate this comment. Although FDA’s MRTP approval process and regulation of reduced harm claims take into account the impact on youth population, they were not designed specifically to address the youth vaping epidemic. Per this comment, they’ve been deleted from lines 87-89 as examples of federal agencies’ response to youth vaping epidemic. The texts have now been revised to “In response to the youth vaping epidemic in the U.S., declared by the Surgeon General in 2018, several federal agencies have taken steps regulating and restricting e-cigarette marketing (1). For example, beginning in April 2018, the U.S. Food and Drug Administration (FDA) and the Federal Trade Commission (FTC) issued multiple warning letters to e-cigarette manufacturers, distributors, and retailers for promoting e-cigarette products in ways misleading to youth or selling e-cigarette products to youth illegally (2-4). In September 2018, the FDA expanded its “The Real Cost” anti-tobacco campaign to target youth vaping and encouraged e-cigarette manufacturers to voluntarily take actions to prevent youth access to their products (5). In November 2018, the FDA Commissioner Gottlieb proposed additional steps to prevent youth access to flavored tobacco products (6).”

Comment #3:

The authors assert on lines 100-101 that “TV remains the most commonly viewed media by youth and adults in the U.S.” with a citation from the Nielsen company dated Quarter 1 2012, almost 9 years ago. Given the rapidly changing media landscape, this assertion should have a more recent citation. This also not a trivial point given the trends that the authors review prior in the section about the increases e-cigarette company direct and indirect digital and social medial advertising driving youth e-cigarette rates. I suggest that the actions of e-cigarette company advertising being directed away from digital/social media back to TV should be contextualized with current information about TV viewership in light of media consumption trends and audience characteristics (with any attendant limitations noted).

Response to Comment #3: This is a great suggestion. As this reviewer correctly pointed out, the media landscape in the U.S. has changed substantially since 2012, more recent data from the 2019 Nielsen reports indicated that in the past decade the amount of time Americans spent on watching TV has steadily declined, and at the same time, the amount of time spent on internet and mobile apps has increased significantly. As of 2019 Q2, the average amount of time Americans adults spent on internet and mobile apps (5:09 hours) has exceeded the average amount of time spent on TV (4:04 hours) per day. This provides important context for understanding e-cigarette company advertising strategies. Absence of federal actions, e-cigarette companies were unlikely to significantly reduce their digital/social media marketing, given the current trend of media consumption. The resurgence of e-cigarettes TV ads could therefore be better understood in the context of these important changes occurred in regulatory environment and media environment. However, TV remains an important media consumption channel for adults, a group purportedly targeted by e-cigarette companies in their 2019 TV ads, which claimed to help adult smokers to switch from combusted cigarettes. We explicitly discussed this in the revised manuscript.

Comment #4:

Currently, the most directly relevant background information is given relatively short shrift in the paragraph on lines 100-113 (which is only 13 of the 71 lines of text in the section). I think space dedicated to this sort of directly relevant background needs to be better balanced with the disproportionately long preceding account of e-cigarette advertising history.

Response to Comment #4: We appreciate this comment. Per this comment, we have shortened the first four paragraphs in the Introduction section and strengthened the most relevant background information. 

Comment #5:

The sentence on lines 63-65 needs a citation.

Response to Comment #5: The citation has been added per this comment.

Comment #6:

On line 130 “Products belonged to the same brand were …” should be edited.

Response to Comment #6: Per this comment, this sentence has been revised as follows: “Products under the same brand were grouped together (e.g., Vuse Alto, Vuse Ciro, Vuse Solo, and Vuse Vibe were all categorized under the Vuse brand).”

Comment #7:

Overall, the approach, data, and methods are sound and consistent with prior studies using similar data that contextualize the results reported here.

Response to Comment #7: We appreciate the positive evaluation of our Materials and methods section by this reviewer. We added more details regarding the data collection process per Reviewer #2’s comments. 

Comment #8:

In the Discussion section, the authors review the changes to e-cigarette company TV advertising in greater detail than in the background section. I think the paper would be stronger if the review and citations related to the various companies’ 2019 adult-focused ad campaigns was located in the background section to properly contextualize the results. For example, the dates of the various campaign launch across the quarters in 2019 correspond with increased TRPs displayed in the results. After informing the reader of these dates at the outset, the authors might consider adding indicator lines on the key quarters of campaign lunch displayed in figures to call readers’ attention to the corresponding results. Some of the less directly relevant literature reviewed in the background can be edited out to focus the section and accommodate the content.

Response to Comment #8: We appreciate this comment. Per this comment, we added the description on e-cigarette companies’ purportedly adult-focused TV ads in 2019 in the Introduction section. We also revised the figures adding indicator lines and texts highlighting the timing of various TV ad campaigns. 

Comment #9:

The manuscript would also be stronger if the authors discussed potential implications of the trends they report beyond the recommendation for continued monitoring. Currently, the conclusions section is quite weak and lessens my enthusiasm for the manuscript. It is important to hear what authors think about their results framed as unintended consequences of FDA scrutiny that motivated changes in e-cigarette company advertising strategies and content. What might the authors expect to see (or not see) if there were continued monitoring? What consequences, for better or worse, do prior studies suggest we may expect from the trends reported here? Some of this is implied in the background and discussion, however the manuscript would be strengthened greatly by spending more space engaging directly in interpretation of results in the discussion section.

Response to Comment #9: We appreciate this comment. We agree with this reviewer that our conclusions section could be strengthened. In our revised manuscript, we addressed this comment by 1) adding more detailed discussion directly interpreting the results from our study, e.g. difference in ad exposure by age group, brand, and media market. 2) discussing the potential implications of the trends with/without additional federal actions restricting e-cigarette marketing. 3) discussing the potential unintended consequences of FDA actions and call for a comprehensive advertising regulatory framework that covers all media channels. 

 

Reviewer #2: 

Comment #1:

This study uses consumer media data to examine the exposure to e-cigarette TV advertising between 2013-2019, and more specifically how advertising has changes in response to tobacco regulation. The paper is well-written. The paper would benefit from a greater interpretation of how the findings may or may not relate to specific tobacco regulations enacted over the past 5 years. The Introduction would benefit from a more explicit description of how this study is unique from existing research. This study covers a longer period of observation than previous studies (2013-2019), however it seems other studies have evaluated advertising exposure by age group and evaluated sales by brand. Why is it necessary to measure marketing exposure when sales data are already available? Also, it would help to frame the reasons for studying differences in marketing exposure by market in the Introduction and interpret the results relevant to the market in the Discussion. As written, this analysis seems out of place. Alternatively, the authors might consider dropping the examination of TV advertising exposure by market from this paper.

Response to Comment #1: We appreciate this comment. Per the comment, we added a more explicit description of the rationale and motivation of our study and the unique contributions of our study in the Introduction section. Our study was motivated by several critical gaps in the current literature: first, although previous studies have document televised e-cigarette advertising in the U.S. up to 2013, no studies have specifically examined televised e-cigarette advertising in the U.S. since 2014, a period when substantial changes occurred in the e-cigarette market, including the emergence of JUUL and other pod-based e-cigarettes, and more recent surge in e-cigarette TV advertising by three major brands since early 2019. Second, previous studies either focused solely on e-cigarette TV ratings or on marketing expenditures, our study presented the data on both TV ratings and TV advertising expenditures. Third, previous studies documented the aggregated e-cigarette TV advertising for the U.S. and did not examine the differences in e-cigarette advertising by media market/geolocation, our study filled these critical gaps. In addition, our study examined e-cigarette advertising by age group and by brand, providing a more rich and detailed understanding of the televised e-cigarette advertising in the U.S.

Although previous studies have documented the sales of e-cigarettes in the U.S., these studies were based on Nielsen scanner data. These data captured sales of e-cigarette in Nielsen participating retail stores and did NOT capture a large portion (over 60%) of e-cigarette sales that occurred in non-participating retailers, online sales, and sales in vape shops and other tobacco specialty stores. In addition, sales data do not reveal marketing exposure among non-users. Consequently, sales should only be used as a supplement, not a substitute for adverting exposure data. 

Per this comment, we added the texts describing the reasons for studying differences in marketing exposure by market in the Introduction. Interpretation of the results relevant to the market has now been added in the Discussion section. We also streamlined the analysis related to market difference.

Comment #2:

Introduction

Line 45- update the high school and middle school e-cig prevalence data to reflect the 2020 estimates.

Response to Comment #2: Per this comment, we updated the e-cigarette prevalence data among high school and middle school students to reflect the 2020 estimates. 

Comment #3:

Lines 78-80- wording of sentence is awkward, consider revising. Is the sentence missing a word?

Response to Comment #3: We appreciate this comment. Per comment #1 and the comment from the other reviewer, this sentence has been removed in the revised manuscript. 

Comment #4:

Methods

Line 120- The Kantar Media tracks TV ads across 20 types of media -- what is meant by 20 types of media? Is TV a type of media (as opposed to print/radio) or is this referring to something different? Please clarify.

Response to Comment #4: The Kantar Media covered a variety of media types, including TV channels (Network TV, Spot TV, Spanish Language Network TV, Cable TV, Syndication, local TV) and Non-TV media platforms (Magazines, Sunday Magazines, Local Magazines, Hispanic Magazines, B-to-B Magazines, National Newspapers, Newspapers, Hispanic Newspapers, Network Radio, National Spot Radio, Local Radio, Local Radio Historical, Internet Display, Mobile Web, Mobile App, Online Video, Mobile Web Video, Internet Search, Outdoor). In this study, we combined the ratings of all TV channels. We clarified this in our revised manuscript. 

Comment #5:

Line 122- Consider adding the list of keywords as a supplementary file.

Response to Comment #5: Per this comment, the list of keywords has been added in S1 Table. 

Comment #6:

Page 7 lines 129-130- Please provide more detail as to the process of "careful review" of e-cigarette products identified by Kantar.

Response to Comment #5: We clarified this “review” process in the revised manuscript. The primary purpose of this review was to identify “false positives,” i.e. product names that contain the prespecified keywords but are not e-cigarette products. For example, “Nike Vapor” contains a keyword “vapor,” but is not an e-cigarette product. As such, advertising data associated with “Nike Vapor” were removed from the search results. Per this comment, this sentence was revised as follows: “E-cigarette products identified from Kantar were carefully reviewed, and irrelevant products, such as “Nike Vapor”, were excluded from data analysis.”.

Comment #7:

Results

Consider condensing the description of Figure 1- focusing on the most salient points. The reader can refer to the figure for details.

Response to Comment #6: Per this comment, we shortened the description of Figure 1 to focus on the most important results, as well as presenting the data that were not explicitly depicted by Figure 1. This paragraph has been revised as follows:

“Fig 1 displays the trends of quarterly e-cigarette TRPs among youth and adults and TV advertising expenditures (in $1,000) from 2013 Q1 to 2019 Q4. Over the study period, e-cigarette advertising TRPs for youth and adults were strongly correlated with the TV advertising expenditures (Pearson correlation = 0.60 for youth and = 0.94 for adults). The e-cigarette advertising TRPs among adults were consistently higher than those of youth. Despite intermittent fluctuations, e-cigarette advertising TRPs and expenditures were relatively stable between 2013 Q1 and 2017 Q1 except for a one-time dip in 2015 Q3. E-cigarette TV advertising were almost non-existent between 2017 Q2 and 2018 Q4, with only a few ads airing during the four quarters from 2017 Q2 to 2018 Q1. Both the youth and adult TRPs increased substantially in 2019 Q1 (from zero ratings to 100.8 among youth and to 528.1 among adults). E-cigarette marketing expenditure also reached to a new high of $12,120,900 in 2019 Q1. Adult TRPs peaked in 2019 Q3 (1,701.9), accompanied by the highest quarterly advertising expenditure ($31,287,100). In 2019 Q4, the expenditure dropped to $6,531,700, and TRPs also declined in both groups (64.9 for youth and 351.6 for adults).”

Comment #8:

The Introduction describes e-cigarette tobacco regulations related to marketing. The figures might benefit from an added overlay identifying when the regulations were enacted. This would allow the reader to easily interpret how regulations may or may not have affected marketing. Table 1 and Figure 1 are redundant-consider moving table 1 to supplementary materials.

Response to Comment #7: This is a great suggestion. Per this comment, we have added an overlay identifying when the regulations were enacted and the timing of key e-cigarette TV ads in Figure 1 and moved Table 1 to the supplementary materials.

Comment #9:

Lines 203-207- The description of how the top brands were identified might be better placed in the Methods.

Response to Comment #8: Per this comment, we have moved the description of how the top brands were identified to the Methods section. 

Comment #10:

Discussion

The first paragraph of the discussion describes how the findings from this analysis are concordant with that previously reported in the literature- consider starting the paragraph with a concise summary of the unique findings from this paper- those perhaps that have not yet been reported in this paper.

Response to Comment #9: Per this comment, we started the first paragraph of the discussion section with a concise summary of the unique findings from our study. It now starts as follows: “Our paper provided important insights into the potential e-cigarette TV advertising exposures among American youth and adults since 2013, a period when significant changes, including the emergence of pod-based e-cigarettes such as JUUL, transformed the U.S. e-cigarette industry and marketplace. We found a resurgence in e-cigarette TV advertising since early 2019, after a lull in 2017/2018. Importantly, this resurgence was driven entirely by three major e-cigarette brands, Blu, JUUL, and Vuse. We also observed significant variations of e-cigarette TV advertising exposure across age groups, media markets, and brands.” 

Comment #11:

Lines 231-238- The description of TV ad expenditure from 2010-2013 seems tangential to the Discussion- would focus more on the paper's findings rather than historical

Response to Comment #10: Per this comment, we shortened the discussion on the historical findings and focused on the study period from 2013 to 2019. 

Comment #12:

Figure 1 shows TV ratings for youth and adults- Is this the TRP? Would use the term TRP throughout the Figures rather than rating if this is the case.

Response to Comment #11: Yes, the ratings were TRPs. We have changed the term to “TV targeted rating points (TRPs)” throughout the Figures in the revised manuscript. 

Comment #13:

Figure 2&3are difficult to read due to many overlying lines- would consider dropping this analysis and Figure from the manuscript or changing the Figures presentation.

Response to Comment #12: Figures 2 & 3 presented important results by media market. We believe it is important to include these figures in the manuscript. Per this comment, we changed the presentation of these Figures to make it easier for readers to identify media market. 

 

Reference

1. Links C. Surgeon General’s Advisory on E-Cigarette Use among Youth. Arizona Free Press: Rockville, MD, USA; 2018.

2. Truth Initiative. E-cigarettes: Facts, stats and regulations 2019 [Available from: https://truthinitiative.org/research-resources/emerging-tobacco-products/e-cigarettes-facts-stats-and-regulations.

3. U.S. Food and Drug Administration. FDA, FTC take action against companies misleading kids with e-liquids that resemble children’s juice boxes, candies and cookies 2018 [Available from: https://www.fda.gov/news-events/press-announcements/fda-ftc-take-action-against-companies-misleading-kids-e-liquids-resemble-childrens-juice-boxes.

4. U.S. Food and Drug Administration. Warning Letters and Civil Money Penalties Issued to Retailers for Selling JUUL and Other E-Cigarettes to Minors 2018 [Available from: https://www.fda.gov/tobacco-products/ctp-newsroom/warning-letters-and-civil-money-penalties-issued-retailers-selling-juul-and-other-e-cigarettes.

5. Rogers K. The FDA Is Launching An Anti-Juul Campaign That’s Like D.A.R.E. for Vaping 2018 [Available from: https://www.vice.com/en_us/article/zm5nw4/the-fda-is-launching-an-anti-juul-campaign-thats-like-dare-for-vaping.

6. U.S. Food and Drug Administration. Statement from FDA Commissioner Scott Gottlieb, M.D., on proposed new steps to protect youth by preventing access to flavored tobacco products and banning menthol in cigarettes 2018 [Available from: https://www.fda.gov/news-events/press-announcements/statement-fda-commissioner-scott-gottlieb-md-proposed-new-steps-protect-youth-preventing-access.

---

## [Decision Letter · Decision Letter 1]

22 Apr 2021

Exposure to e-cigarette TV advertisements among U.S. youth and adults, 2013 – 2019

PONE-D-20-32877R1

Dear Dr. Huang,

We’re pleased to inform you that your manuscript has been judged scientifically suitable for publication and will be formally accepted for publication once it meets all outstanding technical requirements.

Kind regards,

Stanton A. Glantz

Academic Editor

PLOS ONE

Additional Editor Comments (optional):

Reviewers' comments:

Reviewer's Responses to Questions

**Comments to the Author**

1. If the authors have adequately addressed your comments raised in a previous round of review and you feel that this manuscript is now acceptable for publication, you may indicate that here to bypass the “Comments to the Author” section, enter your conflict of interest statement in the “Confidential to Editor” section, and submit your "Accept" recommendation.

Reviewer #1: All comments have been addressed

Reviewer #2: All comments have been addressed

2. Is the manuscript technically sound, and do the data support the conclusions?

Reviewer #1: Yes

Reviewer #2: Yes

3. Has the statistical analysis been performed appropriately and rigorously? 

Reviewer #1: Yes

Reviewer #2: Yes

4. Have the authors made all data underlying the findings in their manuscript fully available?

Reviewer #1: Yes

Reviewer #2: Yes

5. Is the manuscript presented in an intelligible fashion and written in standard English?

Reviewer #1: Yes

Reviewer #2: Yes

6. Review Comments to the Author

Reviewer #1: I appreciate the authors' thoughtful responses to my various comments. Given the various revisions, I believe the manuscript is stronger and there is now enhanced clarity of the value posed by the results.

Reviewer #2: (No Response)

7. PLOS authors have the option to publish the peer review history of their article (what does this mean?). If published, this will include your full peer review and any attached files.

Reviewer #1: No

Reviewer #2: No

---

## [Editor Report · Acceptance letter]

29 Apr 2021

PONE-D-20-32877R1 

Exposure to e-cigarette TV advertisements among U.S. youth and adults, 2013 – 2019 

Dear Dr. Huang:

I'm pleased to inform you that your manuscript has been deemed suitable for publication in PLOS ONE. Congratulations! Your manuscript is now with our production department. 

Kind regards, 

on behalf of

Professor Stanton A. Glantz 

Academic Editor

PLOS ONE